# Sharpness-Aware Minimization Leads to Low-Rank Features

**Maksym Andriushchenko**
EPFL
maksym.andriushchenko@epfl.ch

**Dara Bahri**
Google Research
dbahri@google.com

**Hossein Mobahi**
Google Research
hmobahi@google.com

**Nicolas Flammarion**
EPFL
nicolas.flammarion@epfl.ch

## Abstract

Sharpness-aware minimization (SAM) is a recently proposed method that minimizes the sharpness of the training loss of a neural network. While its generalization improvement is well-known and is the primary motivation, we uncover an additional intriguing effect of SAM: *reduction of the feature rank* which happens at different layers of a neural network. We show that this low-rank effect occurs very broadly: for different architectures such as fully-connected networks, convolutional networks, vision transformers and for different objectives such as regression, classification, language-image contrastive training. To better understand this phenomenon, we provide a mechanistic understanding of how low-rank features arise in a simple two-layer network. We observe that a significant number of activations gets entirely *pruned* by SAM which directly contributes to the rank reduction. We confirm this effect theoretically and check that it can also occur in deep networks, although the overall rank reduction mechanism can be more complex, especially for deep networks with pre-activation skip connections and self-attention layers. We make our code available at https://github.com/tml-epfl/sam-low-rank-features.

## 1 Introduction

Understanding generalization and features learned by overparametrized deep networks is the key topic of modern machine learning. The training objective of deep networks typically has many global optima where the training points are perfectly fitted (Zhang et al., 2017), but very different features and generalization performance are obtained (Chizat et al., 2019; Liu et al., 2019). It has been observed recently that the *sharpness* of a minimum—i.e. how quickly the loss can change in some neighborhood around a minimum in the parameter space—correlates with the generalization error (Keskar et al., 2017; Jiang et al., 2020). The idea of minimizing the sharpness during training to improve generalization has motivated recent works that propose to use worst-case perturbations of the weights on every iteration of training (Foret et al., 2021; Zheng et al., 2021; Wu et al., 2020). We refer to this family of methods collectively as *sharpness-aware minimization* (SAM) and focus on the version proposed in Foret et al. (2021) that uses a single step of normalized gradient ascent to approximate the worst-case weight perturbation.

**Contributions.** In our paper, we are interested in shedding more light on the effect of SAM on the structure of the *features* learned by the network. In particular, our main observation is that:

*Sharpness-aware minimization on overparametrized neural networks leads to low-rank features.*

37th Conference on Neural Information Processing Systems (NeurIPS 2023).

While standard training techniques can already produce features of reduced rank (Huh et al., 2021), SAM significantly amplifies this effect. This effect is of interest when the goal is to identify a low-dimensional structure in the data, as well as for more efficient feature quantization and nearest neighbor retrieval based on the learned features.

We make the following contributions in our paper:

- In Section 3, we present extensive empirical evidence of low-rank features for various models (ResNets, ViTs, MLP-Mixers) trained with SAM on four classification tasks (CIFAR-10/100, Tiny ImageNet, ImageNet-1k) as well as for contrastive text-image training (MS-COCO).
- In Section 4, we provide a mechanistic understanding of how low-rank features arise in a simple two-layer ReLU network. We observe that a significant number of activations gets entirely *pruned* by SAM, which directly contributes to the rank reduction. Furthermore, we provide a theoretical argument supporting this effect of SAM.
- In Section 5, we confirm that this effect can also occur in deep networks, although the overall rank reduction mechanism can be more complex, especially for deep networks with pre-activation skip connections and self-attention layers.
- Finally, in Section 6, we show that directly inducing low-rank features does not improve generalization on natural data. Thus, we conclude that SAM is unique in its ability to both improve generalization *and* decrease the feature rank in a data-adaptive fashion.

## 2 Related work and background knowledge on SAM

In this section, we first present the most relevant previous works and then cover background on SAM.

### 2.1 Related work

**Sharpness-aware minimization.** The idea behind SAM introduced by Foret et al. (2021) is to minimize the sharpness during training to improve generalization. SAM modifies stochastic gradient descent (SGD) such that on every iteration of training, the gradient is taken not at the current iterate but rather at an approximate worst-case point in its vicinity. Zheng et al. (2021) concurrently propose a similar weight perturbation method which also successfully improves standard generalization on multiple deep learning benchmarks. Wu et al. (2020) also propose an almost identical algorithm with the same motivation, but with the focus on improving robust generalization of adversarial training. Chen et al. (2022) discover that SAM is particularly helpful to improve generalization for new architectures like vision transformers (Dosovitskiy et al., 2021) and MLP-Mixers (Tolstikhin et al., 2021). Andriushchenko and Flammarion (2022) study the reasons behind the success of SAM and characterize its effect on simple diagonal linear networks. Bartlett et al. (2022) and Wen et al. (2023) theoretically analyze the regularization effect of SAM close to a minimum.

**Implicit sharpness minimization.** There are multiple works that suggest that sharpness can also be minimized *implicitly* by standard training algorithms. For example, Cohen et al. (2021) formulate the dependency between the learning rate used for training and the sharpness given by the maximum eigenvalue of the Hessian. Barrett and Dherin (2021) and Smith et al. (2021) derive a gradient regularization term that is related to SAM by analyzing SGD and GD with finite step sizes. Damian et al. (2021) suggest a connection between label-noise SGD which implicitly minimizes the trace of the Hessian and SAM whose variations can also minimize the same quantity (Wen et al., 2023). Mulayoff et al. (2021) and Nacson et al. (2023) suggest a relation between sharpness and smoothness for two-layer networks depending on the learning rate of gradient descent.

**Low-rank and sparse features.** The most related work to ours is the one by Ramesh et al. (2022). They briefly note that contrastive language-image pretraining (CLIP) (Radford et al., 2021) with SAM leads to a reduction of the feature rank which they leverage for more efficient feature quantization. Huh et al. (2021) observe that existing training techniques such as SGD on neural networks can already produce features of reduced rank. On a similar note, Ethayarajh (2019) and Cai et al. (2021) observe an extreme anisotropy of the feature space for language models trained with standard methods, including an interesting low-rank cluster structure. Galanti et al. (2022) and Timor et al. (2023) discuss a low-rank effect in the weight matrices due to a small weight norm which is achieved either with weight decay or with a small initialization scale. Na et al. (2022) suggest that models trained with SAM are more compressible via weight pruning and quantization. Gulcehre et al. (2022) observe that

using SAM with dropout can lead to a lower rank in reinforcement learning tasks. Andriushchenko et al. (2023) study the effect of large step size SGD on the Jacobian of the network and activation sparsity. Li et al. (2023) observe that *standard* training leads to extreme activation sparsity in the MLP blocks of transformers.

## 2.2 Background on sharpness-aware minimization

Let $\mathcal{I}_{\text{train}} = \{1, \ldots, n\}$ be the indices of the training set $\{\boldsymbol{x}_i, y_i\}_{i=1}^n$ and $\ell_i(\boldsymbol{\theta})$ be the loss of a model parametrized by weights $\boldsymbol{\theta} \in \mathbb{R}^{|\boldsymbol{\theta}|}$ and evaluated at point $(\boldsymbol{x}_i, y_i)$. Foret et al. (2021) propose to minimize the following worst-case objective instead of standard average loss minimization:

$$\textbf{SAM objective:} \quad \min_{\boldsymbol{\theta} \in \mathbb{R}^{|\boldsymbol{\theta}|}} \mathbb{E}_{\mathcal{I} \subset \mathcal{I}_{\text{train}}} \max_{\|\boldsymbol{\varepsilon}\|_2 \leq \rho} \frac{1}{|\mathcal{I}|} \sum_{i \in \mathcal{I}} \ell_i(\boldsymbol{\theta} + \boldsymbol{\varepsilon}), \tag{1}$$

where $\mathcal{I}$ is a random subset of $m$ training points. We note this objective is based on the maximization of the sum of losses over batches of $m$ points each. To make SAM practical, Foret et al. (2021) propose to minimize the objective (1) with stochastic gradients. Denoting the batch indices at time $t$ by $\mathcal{I}_t$, this leads to the following update rule on each iteration of training:

$$\textbf{SAM update:} \quad \boldsymbol{\theta}_{t+1} := \boldsymbol{\theta}_t - \frac{\gamma_t}{|\mathcal{I}_t|} \sum_{i \in \mathcal{I}_t} \nabla \ell_i \Big( \boldsymbol{\theta}_t + \frac{\rho_t}{|\mathcal{I}_t|} \sum_{j \in \mathcal{I}_t} \nabla \ell_j(\boldsymbol{\theta}_t) \Big). \tag{2}$$

We note that the *same* batch $\mathcal{I}_t$ is used for the inner and outer gradient steps, and $\rho_t$ typically includes the gradient normalization, i.e., $\rho_t := \rho / \|\frac{1}{|\mathcal{I}_t|} \sum_{j \in \mathcal{I}_t} \nabla \ell_j(\boldsymbol{\theta}_t)\|_2$. The worst-case perturbations and the use of a small $m$ in SAM are essential for the generalization improvement which depends continuously on $m$ as noted in Foret et al. (2021).

# 3 Experimental evidence of low-rank features due to SAM

We first discuss how we measure the feature rank and then present strong empirical evidence for low-rank features for models trained with SAM. We consider first classification tasks on CIFAR-10, CIFAR-100 (Krizhevsky and Hinton, 2009), Tiny ImageNet (Le and Yang, 2015), and ImageNet-1k (Deng et al., 2009), and then contrastive learning on MS-COCO (Lin et al., 2014).

**How we measure the feature rank.** Consider a neural network $f : \mathcal{X} \to \mathbb{R}^D$ which maps inputs from a set $\mathcal{X}$ (e.g., images) to a $D$-dimensional vector (e.g., logits for classification tasks or embeddings for contrastive learning). We assume that $f$ can be decomposed on $B$ *blocks*, i.e., $f(\boldsymbol{x}) = f_B \circ f_{B-1} \circ \ldots \circ f_1(\boldsymbol{x})$ where a single block $f_b : \mathbb{R}^{d_{b-1}} \to \mathbb{R}^{d_b}$ can consist of multiple layers. By *features* we refer to the intermediate values $f_b(\boldsymbol{x}) \circ \ldots \circ f_1(\boldsymbol{x}) \in \mathbb{R}^{d_b}$ at some block $b$. To assess their rank, we first do a PCA on the feature matrix $[f_b(\boldsymbol{x}_i)^\top]_{i=1}^{d_b} \in \mathbb{R}^{n \times d_b}$ and then select *the minimal number of principal components that span* $99\%$ *of the variance in the data*. We refer to this measure as the feature rank, as it captures the notion of the most informative features in an interpretable manner. We note that this is not equivalent to the matrix rank in a strict mathematical sense. However, it provides a practical alternative to selecting a fixed threshold on singular values, which can often be challenging to determine.

## 3.1 Low-rank features for ResNets on standard classification tasks

**Setup.** We train a PreAct ResNet-18 (He et al., 2016b) with standard augmentations on standard deep learning datasets: CIFAR-10, CIFAR-100, and Tiny ImageNet. We consider both (1) a *minimal setting* with a small learning rate, no momentum, and no weight decay, and (2) a *state-of-the-art setting* which includes large learning rates, momentum, and weight decay. Since we observe that *neural collapse*—convergence of the feature rank of the penultimate layer to the number of classes (Papyan et al., 2020)—occurs in a cascading fashion (Hui et al., 2022) and interferes with the low-rank trend of SAM, we exclude the last residual superblock and instead report the rank at the *third* superblock.[1]

**Observations.** We plot the main metrics for these three datasets in Figure 1, 2, and 3. We observe that the models trained with SAM achieve a *substantially* smaller feature rank which occurs not

---

[1]Further details can be found in our code https://github.com/tml-epfl/sam-low-rank-features.

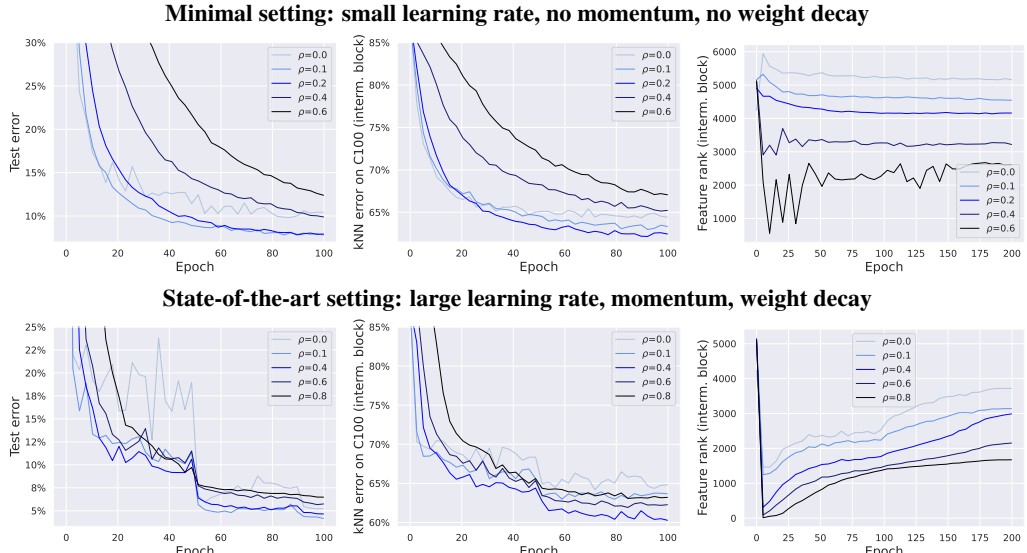

**Figure 1: ResNet-18 on CIFAR-10.** SAM improves test error (*left*), leads to more generalizable features (*middle*), and noticeably reduces the feature rank at the intermediate ResNet block (*right*). Note that the test error improvement is U-shaped in $\rho$ unlike the rank reduction which is monotonic.

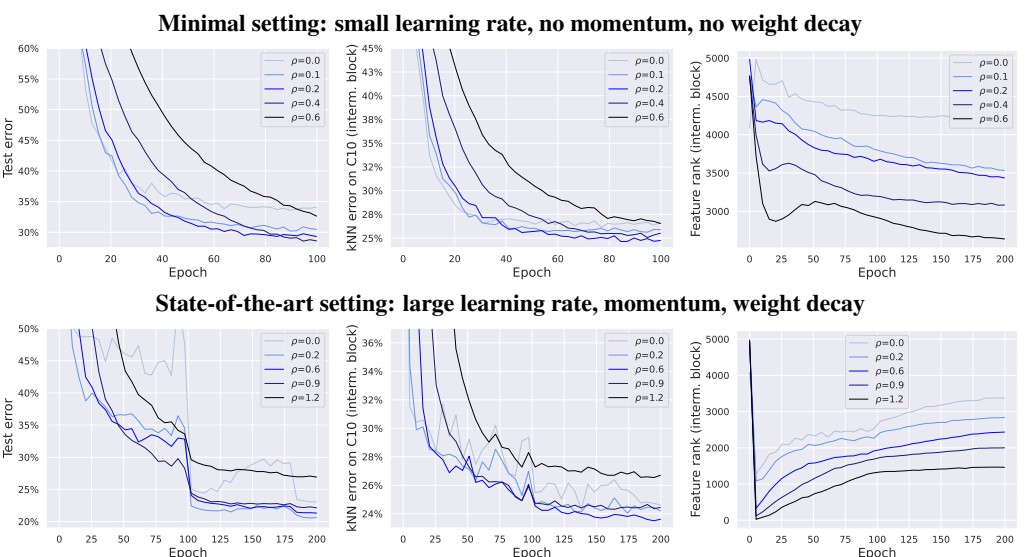

**Figure 2: ResNet-18 on CIFAR-100.** SAM improves test error (*left*), leads to more generalizable features (*middle*), and noticeably reduces the feature rank at the intermediate ResNet block (*right*). Note that the test error improvement is U-shaped in $\rho$ unlike the rank reduction which is monotonic.

only at the end but also *throughout* the training. We also note that the generalization improvement is *U-shaped* in $\rho$ (i.e., too large $\rho$ is harmful) unlike the rank reduction which is monotonic. We note that for the state-of-the-art setting, the rank exhibits a sudden drop at the beginning and then a gradual increase, both for standard training and SAM. We verified that such behavior originates from the usage of initial large learning rates and is not specific to SAM. Overall, the rank reduction effect is very consistent over the three datasets and over both minimal and state-of-the-art settings. We also observe that augmentations play a crucial role in revealing the low-rank trend with respect to the $\rho$ of SAM, whereas the addition of only weight decay is insufficient for revealing a similar trend. Additionally, to confirm the generalizability of the low-rank features taken *at an intermediate layer*, we also include transfer learning performance using k-nearest neighbors classification with $k = 10$. For this purpose, we compute (1) the k-NN classification error on CIFAR-10 for models trained on

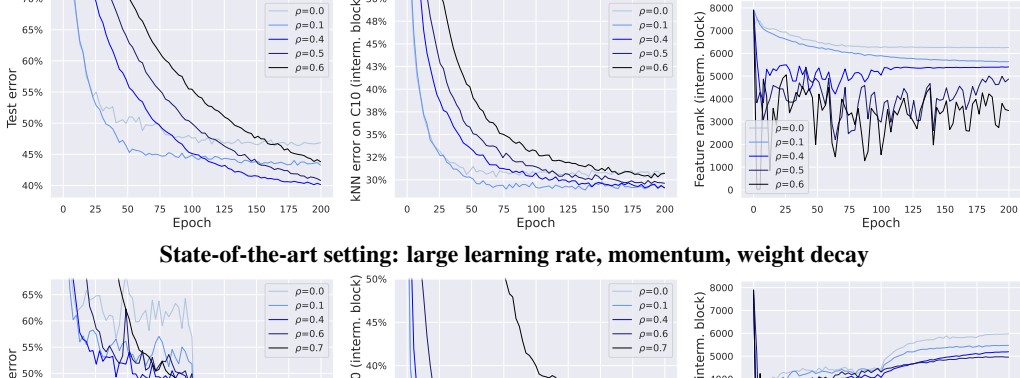

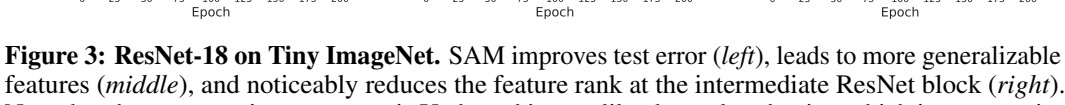

**Figure 3: ResNet-18 on Tiny ImageNet.** SAM improves test error (*left*), leads to more generalizable features (*middle*), and noticeably reduces the feature rank at the intermediate ResNet block (*right*). Note that the test error improvement is U-shaped in $\rho$ unlike the rank reduction which is monotonic.

CIFAR-100 and (2) the k-NN classification error on CIFAR-100 for the models trained on CIFAR-10 and Tiny ImageNet. Without this experiment, one could assume that since these features are not from the penultimate layer, they can be of limited use for downstream tasks. However, this experiment highlights that it is not the case: the block-3 features are still suitable for transfer learning and show completely non-trivial performance.

**Additional experiments.** We report a few more related experiments in Appendix B. First, we note that the feature rank shows the same trend on both training and test sets and is not sensitive to the chosen variance threshold (see Figure 9 and 10 in Appendix). Moreover, the same picture holds also for different batch sizes (see Figure 11 in Appendix). Additionally, we also report results for more recent variants of SAM such as ASAM from Kwon et al. (2021) and GAM from Zhang et al. (2023) (see Table 2 in Appendix). Finally, we discuss the behavior of the feature rank at the *penultimate* layer instead of the intermediate ResNet block (see Figure 12 in Appendix).

### 3.2 Low-rank features for ViTs and MLP-Mixers on ImageNet-1k

**Setup.** We take publicly available models trained on ImageNet-1k from the `vision_transformer` library[2] which contains models from Dosovitskiy et al. (2021), Tolstikhin et al. (2021), and Chen et al. (2022). We evaluate the feature rank on $12\,800$ examples using the zeroth token for ViTs and average features over all tokens for MLP-Mixers.

**Observations.** We report the results in Table 1. We note that neural collapse is not an issue here since the feature dimension is smaller than the number of classes on ImageNet-1k. We see a very consistent rank reduction trend from SAM for each setting, with the only exception of MLP-Mixers after $3/4$ of the total number of blocks. Finally, we note that these models are trained with rather small $\rho$ of SAM (e.g., $0.2$ for ViT-B/16), and we can expect a stronger low-rank effect for higher $\rho$.

### 3.3 Low-rank features in contrastive language-image training on MS-COCO

**Setup.** Here we use a training setting similar to CLIP (Radford et al., 2021) but we start from pretrained image and language models. We fine-tune the R+Ti/16 vision transformer from Steiner et al. (2021) and BERT from Devlin et al. (2018) on MS-COCO using the InfoNCE contrastive loss (Oord et al., 2018). We add a linear head to each of the encoders to match the dimension of both pre-trained encoders. We note that for contrastive training, unlike for classification, neural collapse is

---

[2] https://github.com/google-research/vision_transformer

**Table 1: ViTs and MLP-Mixers on ImageNet-1k.** We compute the feature ranks for publicly available models from the `vision_transformer` library.

| Training | Block | ViT-B/16 | ViT-B/32 | ViT-L/16 | ViT-L/16 | Mixer-B/16 |
|----------|-------|----------|----------|----------|----------|------------|
| Standard | Last | 680 | 672 | 903 | 898 | 695 |
| SAM | Last | **617** | **654** | **820** | **844** | **620** |
| Standard | After $^3/_4$ blocks | 467 | 484 | 595 | 595 | **301** |
| SAM | After $^3/_4$ blocks | **426** | **440** | **314** | **442** | 477 |
| Standard | After $^1/_2$ blocks | 412 | 390 | 387 | 469 | 425 |
| SAM | After $^1/_2$ blocks | **346** | **362** | **250** | **318** | **369** |

**Figure 4: Contrastive image-text training on MS-COCO.** SAM improves the retrieval error (*top*) and reduces the feature rank at the last layer of both image and text encoders (*bottom*).

not an issue, so we report the feature rank directly at the last layer. We measure the retrieval error within each batch of size $128$.

**Observations.** In Figure 4, we observe that SAM both substantially improves the retrieval error and leads to features of much smaller rank. We note that having features of reduced rank may contradict the common intuition that low rank is typically harmful in contrastive learning (Chen and He, 2021; Hua et al., 2021), however, we are still in the regime that the remaining dimensions are expressive enough to preserve (and even improve) the retrieval performance. In addition, we note that the low-rank features are observed both in the image *and* text encoders suggesting that this effect of SAM is not specific to image data. Moreover, we observe the low-rank effect also when the text encoder is frozen during fine-tuning (Figure 13 in Appendix). Thus, even a single linear layer on the side of text features can be sufficient to get the low-rank effect.

## 4 How SAM can induce low-rank features: insights from simple models

In this section, we dissect the source of the low-rank effect of SAM on two-layer neural networks. We first provide clear empirical evidence that this effect can be attributed to zero activations and then confirm it theoretically.

### 4.1 Mechanistic understanding of low-rank features for a two-layer ReLU network

**Setup.** We consider a ReLU network $f(\boldsymbol{\theta}) = \langle \boldsymbol{a}, \sigma(\boldsymbol{W}\boldsymbol{x}) \rangle$ trained with the squared loss and parametrized by $\boldsymbol{\theta} = [\text{vec}(\boldsymbol{W}), \boldsymbol{a}]$ where $\boldsymbol{W} \in \mathbb{R}^{m \times d}$ and $\boldsymbol{a} \in \mathbb{R}^m$. We use the teacher-student

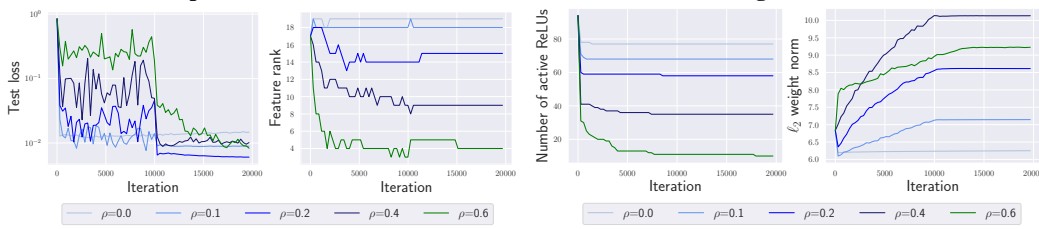

**Figure 5: Two-layer ReLU networks in the teacher-student setup.** The models trained with a higher $\rho$ of SAM generalize better, have a significantly smaller feature rank, smaller number of active ReLUs, and higher $\ell_2$ weight norm. Note that the generalization improvement is U-shaped in $\rho$ unlike the rank reduction which is monotonic.

setup with 3 teacher neurons, $m = 100$ student neurons, and input dimension $d = 3$ inspired by the setup of Chizat et al. (2019). The goal of the student network is to recover the teacher network from a finite set of training points which are sampled from the Gaussian distribution and labeled by the teacher network. We use SAM for the first $50\%$ of iterations and then disable it to achieve full convergence as we observed that running SAM, especially with high $\rho$, hinders convergence. To obtain a smooth trend of rank reduction and clear separation between different $\rho$, we exceptionally use the PCA variance threshold of $99.99\%$, which is higher than the $99\%$ used in all other experiments.

**Low-rank features are due to zero activations.** We provide a mechanistic understanding of how low-rank features emerge in Figure 5. First, we observe that even in this simple setting, SAM improves generalization and learns features of substantially smaller rank (4 instead of 19). We investigate the origin of low-rank features by examining the number of active ReLU units, which noticeably decreases with increasing $\rho$. Although sparse activations do not necessarily imply low-rank features (e.g., the identity matrix is sparse but full-rank), we observe that SAM sets entire activations to zero during training. For the largest $\rho = 0.6$, only 10 out of 100 neurons are activated on at least one training example, and even fewer of them contribute to the $99.99\%$ PCA variance in the data. Similar results are shown for other activations, such as tanh and absolute value (see Figure 14 in Appendix), where the low-rank effect can be similarly attributed to zero activations.

**SAM $\neq$ weight decay.** Interestingly, SAM reduces the number of active ReLUs not by shrinking to zero the weight vectors associated with ReLU units, but by leveraging the negative part of ReLU to effectively prune neurons. As suggested in Figure 5 (right), the weight norms only increase with $\rho$, and the effect of SAM is *opposite* to that of weight decay. Thus, theoretical results demonstrating that weight decay leads to low-rank features (Galanti et al., 2022; Timor et al., 2023) are not directly applicable to studying the low-rank effect of SAM. This observation highlights the difference between SAM and classical regularization methods and explains why we cannot achieve the same regularization effect with weight decay. To better understand this phenomenon for this simple model, we need to find a theoretical argument *specific to SAM*.

### 4.2 Theoretical argument for low-rank features in a two-layer ReLU network

Let $\ell(\boldsymbol{\theta})$ be the loss on example $(\boldsymbol{x}, y)$ sampled by SGD and assume the squared loss for concreteness. We formulate our theoretical argument in the following proposition.

**Proposition 1.** *Every update of SAM contains a component that decreases all pre-activation values $\{\langle w_j, x \rangle\}_{j=1}^m$ of a two-layer ReLU network trained with the squared loss by a non-negative amount equal to $\eta\rho\sqrt{\ell(\boldsymbol{\theta})}/\|\nabla f(\boldsymbol{\theta})\|_2\sigma(\langle \boldsymbol{w}_j, \boldsymbol{x} \rangle)\|\boldsymbol{x}\|_2^2$.*

*Proof.* We have for the update of SAM:

$$\nabla\ell\left(\boldsymbol{\theta} + \rho\frac{\nabla\ell(\boldsymbol{\theta})}{\|\nabla\ell(\boldsymbol{\theta})\|_2}\right) = \nabla\ell(\boldsymbol{\theta}) + \rho\nabla^2\ell(\boldsymbol{\theta})\frac{\nabla\ell(\boldsymbol{\theta})}{\|\nabla\ell(\boldsymbol{\theta})\|_2} + \mathcal{O}(\rho^2) = \nabla\left[\ell(\boldsymbol{\theta}) + \rho\|\nabla\ell(\boldsymbol{\theta})\|_2 + \mathcal{O}(\rho^2)\right].$$

Thus, under the first-order approximation, a step of SAM corresponds to a gradient update on the regularized objective $\ell(\boldsymbol{\theta}) + \rho\|\nabla\ell(\boldsymbol{\theta})\|_2$. Now recall that the layerwise gradients of a two-layer

ReLU network can be written as:

$$\nabla_{\boldsymbol{a}}\ell(\boldsymbol{\theta}) = \ell'(\boldsymbol{\theta}) \cdot \sigma(\boldsymbol{W}\boldsymbol{x}), \quad \nabla_{\boldsymbol{w}_j}\ell(\boldsymbol{\theta}) = \ell'(\boldsymbol{\theta}) \cdot a_j \sigma'(\langle \boldsymbol{w}_j, \boldsymbol{x}\rangle)\boldsymbol{x}.$$

Then a direct computation gives us the following expression for the full gradient norm:

$$\|\nabla\ell(\boldsymbol{\theta})\|_2 = |r| \cdot \|\nabla f(\boldsymbol{\theta})\|_2 = |\langle \boldsymbol{a}, \sigma(\boldsymbol{W}\boldsymbol{x})\rangle - y|\sqrt{\|\sigma(\boldsymbol{W}\boldsymbol{x})\|_2^2 + \|\boldsymbol{x}\|_2^2 \cdot \|\boldsymbol{a} \odot \sigma'(\boldsymbol{W}\boldsymbol{x})\|_2^2},$$

where $\odot$ denotes element-wise multiplication and $r$ denotes the residual $f(\boldsymbol{\theta}) - y$. Then the update of $\boldsymbol{w}_j$ for neuron $j$ on each step of SAM with step size $\eta$ can be written as:

$$\boldsymbol{w}_j := \boldsymbol{w}_j - \eta(\nabla\ell(\boldsymbol{\theta}) + \rho\nabla\|\nabla\ell(\boldsymbol{\theta})\|_2) + \mathcal{O}(\rho^2)$$

$$\boldsymbol{w}_j := \boldsymbol{w}_j - \underbrace{\eta r \left(1 + \rho\|\nabla f(\boldsymbol{\theta})\|_2/\sqrt{\ell(\boldsymbol{\theta})}\right) a_j \sigma'(\langle \boldsymbol{w}_j, \boldsymbol{x}\rangle)\boldsymbol{x}}_{\text{data fitting term}} - \underbrace{\eta\rho\sqrt{\ell(\boldsymbol{\theta})}/\|\nabla f(\boldsymbol{\theta})\|_2 \sigma(\langle \boldsymbol{w}_j, \boldsymbol{x}\rangle)\boldsymbol{x}}_{\text{regularization component}} + \mathcal{O}(\rho^2)$$

where we used the fact that $\sigma'(\langle \boldsymbol{w}_j, \boldsymbol{x}\rangle)\sigma(\langle \boldsymbol{w}_j, \boldsymbol{x}\rangle) = \sigma(\langle \boldsymbol{w}_j, \boldsymbol{x}\rangle)$ and second-order terms are zero almost everywhere for ReLUs. The data fitting term is the same as normal gradient but with a larger effective learning rate $\eta(1 + \rho\|\nabla f(\boldsymbol{\theta})\|_2/\sqrt{\ell(\boldsymbol{\theta})})$ instead of $\eta$. The most interesting term is the one coming from $\nabla\|\nabla f(\boldsymbol{\theta})\|_2$ which drives pre-activations $\langle \boldsymbol{w}_j, \boldsymbol{x}\rangle$ to negative values *on every step of SAM* which can be seen directly from the update rule:

$$\langle \boldsymbol{w}_j, \boldsymbol{x}\rangle := \langle \boldsymbol{w}_j, \boldsymbol{x}\rangle - \eta r \left(1 + \rho\|\nabla f(\boldsymbol{\theta})\|_2/\sqrt{\ell(\boldsymbol{\theta})}\right) a_j \sigma'(\langle \boldsymbol{w}_j, \boldsymbol{x}\rangle)\|\boldsymbol{x}\|_2^2$$
$$- \eta\rho\sqrt{\ell(\boldsymbol{\theta})}/\|\nabla f(\boldsymbol{\theta})\|_2 \sigma(\langle \boldsymbol{w}_j, \boldsymbol{x}\rangle)\|\boldsymbol{x}\|_2^2 + \mathcal{O}(\rho^2),$$

where we note that $\eta\rho\sqrt{\ell(\boldsymbol{\theta})}/\|\nabla f(\boldsymbol{\theta})\|_2 \sigma(\langle \boldsymbol{w}_j, \boldsymbol{x}\rangle)\|\boldsymbol{x}\|_2^2$ is always non-negative for ReLU. □

We confirm empirically in Figure 15 in Appendix that using only the first-order terms in the SAM update leads to the same effect in all key metrics including the feature rank and the number of active ReLUs. Overall, this mechanism suggests how SAM can *suppress redundant activations* if they are not needed to fit the training data. Moreover, this effect takes place at every iteration of SGD but is stronger at the beginning of training since it is proportional to the loss $\sqrt{\ell(\boldsymbol{\theta})}$. Moreover, a similar argument can be made for a multi-layer network which can explain why the low-rank effect occurs at *multiple* layers since the $\|\nabla f(\boldsymbol{\theta})\|_2$ term has activations of all layers in it. Also it suggests an intuitive picture of what a flat minimum of SAM means in terms of the learned function: a minimum that corresponds to a network sparsely activated on the training data.

## 5 Investigation of low-rank mechanisms on deep networks

In this section, we confirm that the low-rank mechanism described above can also occur in post-activation ResNets, although the overall rank reduction mechanism can be more complex, particularly for networks with pre-activation skip connections and self-attention layers.

**Post-activation ResNet on CIFAR-10.** We consider a standard, *post-activation* ResNet-18 (He et al., 2016a) with 4 residual superblocks trained on CIFAR-10 in the minimal setting (i.e., small step sizes, no momentum, no weight decay). Since values in the residual stream of this network (unlike for PreAct ResNets) are taken *after* ReLU, we can expect to see the ReLU pruning effect described in the previous section. We count a ReLU as *active* if at least on one training example, it achieves at least 5% of the maximum value in the feature matrix. We plot the results in Figure 6 which confirms the ReLU pruning effect: the number of active ReLUs decreases with $\rho$ at later residual blocks. For example, for $\rho = 0.6$, the number of active ReLUs at block 4 reaches 77%, which directly contributes to a lower feature rank, compared to 100% of standard training. Finally, we note that this mechanism is not likely to be the only one contributing to the lower rank since the trends for the feature rank and number of active ReLUs do not perfectly agree.

**Pre-activation ViT on MS-COCO.** Next, we consider the ViT with 12 *pre-activation* residual blocks used in our experiments on MS-COCO. First, we show how the feature ranks change in the *residual stream* after attention and MLP subblocks in Figure 7 (*left, middle*). We observe that almost each attention block gradually increases the feature rank, *but the increase is smaller for models with higher $\rho$*. At the same time, MLP blocks can both increase and decrease the rank, but for higher $\rho$, the difference between post-MLP rank and pre-MLP rank tends to be larger. Thus, we conclude that the

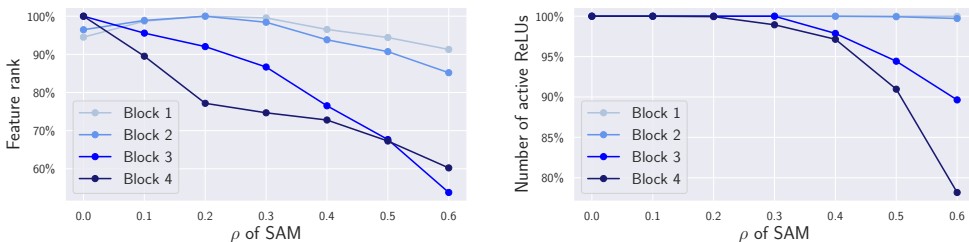

**Figure 6: Post-activation ResNet-18 on CIFAR-10.** Both the feature rank and the number of active ReLUs tend to decrease with the $\rho$ of SAM, particularly at later ResNet blocks.

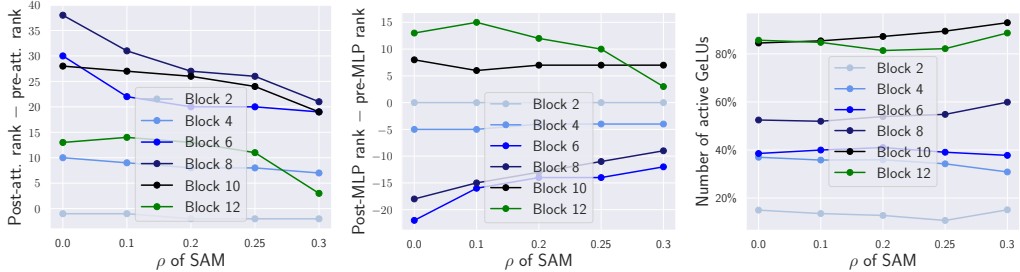

**Figure 7: ViT image encoder on MS-COCO.** We show the difference in the feature ranks before and after attention subblocks (*left*), before and after MLP subblocks (*middle*), and the number of active GeLUs inside of residual branches (*right*).

rank reduction effect is coming primarily from the attention blocks. Moreover, we do not observe any coordinate-aligned sparsity in the residual stream. Additionally, we plot the number of active GeLUs inside of MLP subblocks in Figure 7 (*right*). Since GeLU can be seen as a differentiable approximation of ReLU, the analogous notion of active GeLU units also makes sense. However, we do not observe any decrease in the number of GeLUs over the $\rho$ of SAM for these models. We show more detailed plots about the behavior of feature ranks in Figure 16 in Appendix. Moreover, in Figure 17 we show that even when the text encoder remains frozen and only a single linear layer is trained on top of it, the low-rank effect is still observed. This indicates that SAM can effectively utilize even a *single matrix multiplication* to achieve features of reduced rank. Overall, we conclude that the mechanism behind the low-rank effect of SAM can vary significantly, and SAM can leverage different components of the network, including activations, attention layers, and even individual linear layers.

## 6 Do low-rank features lead to better generalization on natural data?

We conclude the discussion on the low-rank features of SAM by demonstrating that directly inducing low-rank features *does not* result in improved generalization for natural data such as MS-COCO.

**Setup.** We induce the low-rank directly at the last layer by using a *linear bottleneck layer* which is also known as the *Burer-Monteiro factorization* (Burer and Monteiro, 2003): we parametrize the weight matrix $W^{(L)} \in \mathbb{R}^{d_{L-1} \times d_L}$ at the last layer $L$ as $W^{(L)} = U^{(L)}V^{(L)}$ where $U^{(L)} \in \mathbb{R}^{d_{L-1} \times h}$ and $V^{(L)} \in \mathbb{R}^{h \times d_L}$. We perform training on MS-COCO by varying the inner dimension $h$ for the last layers of both image and text encoders, while keeping the remaining training process the same as described in Section 3.

**Observations.** We plot the results in Figure 8 where we observe that enforcing low-rank features alone *does not* consistently enhance generalization. Introducing bottleneck layers only marginally improves the imagewise retrieval error (from 23.3% to 22.7%) but negatively affects textwise retrieval (from 22.2% to 22.9%). In contrast, SAM demonstrates a significant improvement of $-4.7\%$ and $-3.0\%$ in the respective retrieval tasks. Thus, we conclude that SAM stands out in its ability to simultaneously enhance generalization and select low-rank features in a data-adaptive manner.

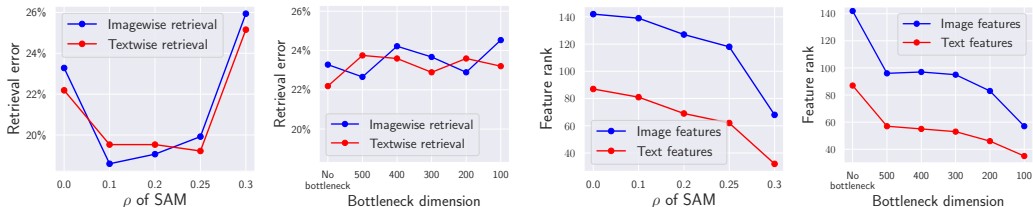

**Figure 8: Contrastive image-text training on MS-COCO.** We compare the effect of SAM and linear bottleneck layers on the retrieval error and feature ranks of the last layer. We observe that linear bottleneck layers do not improve the retrieval error at all, unlike SAM.

## 7 Discussion and future directions

Finally, we discuss the implications of low-rank features learned by SAM as well as future directions.

**More efficient retrieval.** As illustrated in Section 3, despite the lower rank of the features learned by SAM, their effectiveness for *nearest neighbor retrieval* is preserved or even improved. This suggests that one can achieve faster retrieval by reducing the dimensionality of the feature space by using, for example, only the top-$k$ components of PCA. Furthermore, the low-rank bias of SAM appears to be *data-adaptive*, meaning that it adjusts to the data structure rather than relying on a fixed predetermined dimension.

**More efficient feature quantization.** On a related note, having features of reduced rank can also be helpful to speed up *feature quantization*. This is confirmed, in particular, in the DALL·E 2 paper (Ramesh et al., 2022) which reports that this effect combined with PCA can lead to a $3\times$ reduction in the number of tokens predicted during inference, and also can improve the training stability.

**Understanding the features learned by SAM.** Developing a better understanding of the features learned by popular methods such as SAM is an important goal on its own. The low-rank feature bias can be advantageous in scenarios where the data is generated by a low-rank function. However, for realistic datasets like MS-COCO, the impact of this effect on generalization is not as significant, yet it still remains a valuable side effect.

**Future directions.** Our findings suggest that low-rank features alone do not enhance generalization for natural data beyond simple teacher-student setups. Consequently, the low-rank effect of SAM appears to be a useful *side effect*. Therefore, understanding the impact of SAM on learned features that leads to generalization improvements on natural data remains an open question. Additionally, further theoretical analysis of the low-rank effect of SAM for more complex architectures, such as those involving skip-connections and self-attention layers, would be of great interest.

## Acknowledgements

We thank the anonymous reviewers for their suggestions that helped to improve the paper. The work is supported by the Google/EPFL Research Collab program. M.A. was additionally supported by the Google Fellowship and Open Phil AI Fellowship.

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

# Appendix

The appendix is organized as follows:

- Section A: implementation details and hyperparameters of all experiments.
- Section B: additional results for classification experiments on CIFAR-10, CIFAR-100, and Tiny ImageNet.
- Section C: additional results for contrastive image-text training on MS-COCO.
- Section D: additional results for two-layer networks.
- Section E: additional results for investigations of the low-rank mechanism on deep networks.

## A   Experimental details

In this section, we specify the implementation details and hyperparameters of all our experiments.

**Experiments on CIFAR-10, CIFAR-100, Tiny ImageNet.** We train pre-activation ResNets-18 (He et al., 2016b) on CIFAR-10, CIFAR-100, Tiny ImageNet for experiments in Section 3 and post-activation ResNets-18 for experiments in Section 5. We train these models with batch size 256 for 200 epochs using standard augmentations (random crops and random mirroring). For the minimal setting, we use plain SGD with the learning rate 0.05. For the state-of-the-art setting, we use SGD with the learning rate 0.1 (decayed by a factor of $10\times$ after $50\%$ and $90\%$ epochs), momentum parameter 0.9, weight decay 0.0005. We train models with a grid of different parameters $\rho$ of SAM where we adjust the grid such that it includes both values of $\rho$ that improve generalization (small $\rho$, usually in the range $[0.05, 0.2]$) but also those which slightly degrade it (large $\rho$, usually larger than 0.5). We evaluate the feature rank on $10\,240$ training examples by counting the minimal number of PCA components needed to span $99\%$ variance.

**Experiments on ImageNet-1k.** We refer to the original papers Dosovitskiy et al. (2021), Tolstikhin et al. (2021), and Chen et al. (2022) for the details on how these models were trained. We evaluate the feature rank on $12\,800$ training examples by counting the minimal number of PCA components needed to span $99\%$ variance.

**Experiments on MS-COCO.** We fine-tune the R+Ti/16 vision transformer from Steiner et al. (2021) and BERT from Devlin et al. (2018) on MS-COCO using the InfoNCE contrastive loss. We add a linear head to each of the encoders with the final feature dimension of 768. We use Adam (Kingma and Ba, 2014) with learning rate 0.0001 which is decayed down to 0 using a cosine decay schedule. We train these models with batch size 128 for 25 epochs without data augmentations. We evaluate the feature rank on $1\,280$ training examples by counting the minimal number of PCA components needed to span $99\%$ variance. To aggregate the features over different tokens, we always take the embedding of the first token since this is what is used for classification of the pretrained model.

**Experiments on two-layer networks in the teacher-student setup.** We use the teacher-student setup with 3 teacher neurons, $m = 100$ student neurons, and input dimension $d = 3$. We consider training with stochastic gradients based on mini batch size of 1 using the learning rate of 0.1. We initialize the weights of the teacher network with the standard deviation of 1.0 and the student network with the standard deviation of 0.3. We use SAM for the first $50\%$ of iterations and then disable it to achieve full convergence as we observed that running SAM, especially with high $\rho$, hinders convergence. We evaluate the feature rank on all 20 training examples by counting the minimal number of PCA components needed to span $99.99\%$ variance. We exceptionally choose this PCA threshold (instead of $99\%$ as in all other experiments) to obtain a smooth trend of rank reduction and clear separation between different $\rho$.

**Computational resources.** We performed all experiments on a single Nvidia A100 GPU where we used an internal cluster for all experiments except experiments on MS-COCO, for which we used a cloud provider. A typical training run on CIFAR-10 and CIFAR-100 takes around 2 hours, on Tiny ImageNet around 5 hours, and on MS-COCO around 7 hours.

# B  Additional results for classification

In Section 3, we focused on the *original* SAM as introduced in Foret et al. (2021) since it is still the most popular SAM variant in the community and it is implemented without any further approximations. However, to provide a comprehensive comparison of SAM variants, we also include results on more recent variants of SAM such as ASAM (Adaptive Sharpness-Aware Minimization) (Kwon et al., 2021) and GAM (Gradient norm Aware Minimization) (Zhang et al., 2023). Similarly to our main experiments in Section 3, we also use ResNets on CIFAR-10. We select the default settings given in their code repositories (which includes a smaller ResNet for ASAM compared to GAM) and vary only the perturbation radius $\rho$. We report the results in Table 2 that confirm that the low-rank observation also extends to other recent SAM variants.

| $\rho$ **of ASAM** | 0.0 | 0.5 | 1.0 | 2.0 | 4.0 |
|---|---|---|---|---|---|
| **Test error** | 7.29% | 6.53% | 6.38% | 7.12% | 10.64% |
| **Feature rank** | 5048 | 4801 | 4699 | 4578 | 4383 |

| $\rho$ **of GAM** | 0.0 | 0.2 | 0.4 | 0.8 | 1.6 |
|---|---|---|---|---|---|
| **Test error** | 4.04% | 3.65% | 3.64% | 3.81% | 4.81% |
| **Feature rank** | 7633 | 7381 | 7303 | 6897 | 6927 |

**Table 2:** Results for more recent variants of SAM: ASAM (Kwon et al., 2021) and GAM (Zhang et al., 2023).

In Figure 9, we plot feature ranks computed on the training and test sets of CIFAR-10 and CIFAR-100 for post-activation and pre-activation ResNets-18, respectively, taken at the last epoch. We select two different architectures to check the generality of our findings. We observe that the feature ranks stay very close, no matter if we evaluate them on the training or test sets. Thus, we conclude that the low-rank effect of SAM generalizes also to unseen samples.

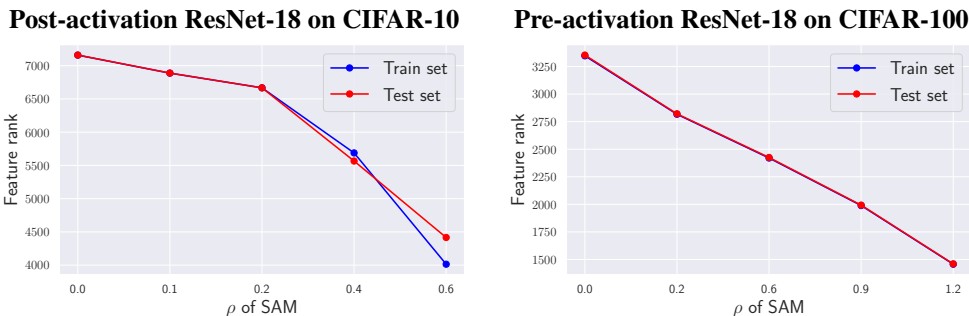

**Figure 9: ResNets-18 on CIFAR-10 and CIFAR-100.** Feature ranks computed on the training and test sets are very similar to each other. We conclude that the low-rank effect of SAM generalizes also to unseen samples.

In Figure 10, we plot feature ranks for 95%, 99%, 99.9% PCA variance. Feature ranks computed with different PCA variance thresholds exhibit very similar trends. We observe that the overall trend and relative ranking between models trained with different $\rho$ is unaffected by the choice of the PCA variance threshold.

In Figure 11, we plot metrics like in Figure 1 but for larger batch sizes (512 and 1024 instead of 256). We see that the low-rank trend is clearly visible there as well.

Finally, in Figure 12, we show feature ranks computed at the *last* block of ResNet (just before the linear head). We see that the neural collapse phenomenon occurs: the feature rank converges to 10 (i.e., the number of classes) even for $\rho = 0$. The feature rank with SAM is lower at the beginning but then increases to a slightly higher value than 10 towards the end of training.

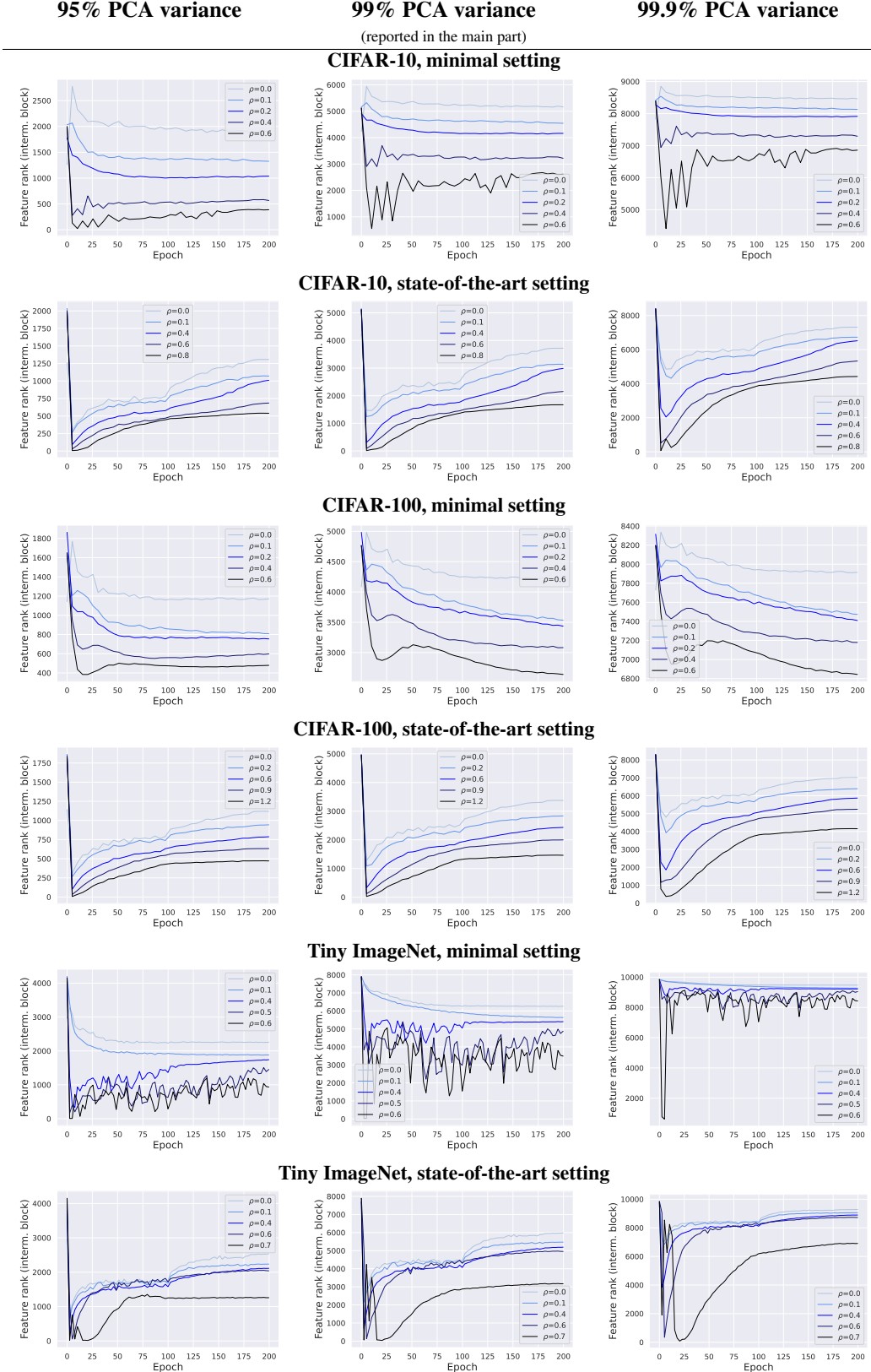

**Figure 10: ResNet18 on CIFAR-10, CIFAR-100, Tiny ImageNet.** Feature ranks computed with different PCA variance thresholds exhibit very similar trends.

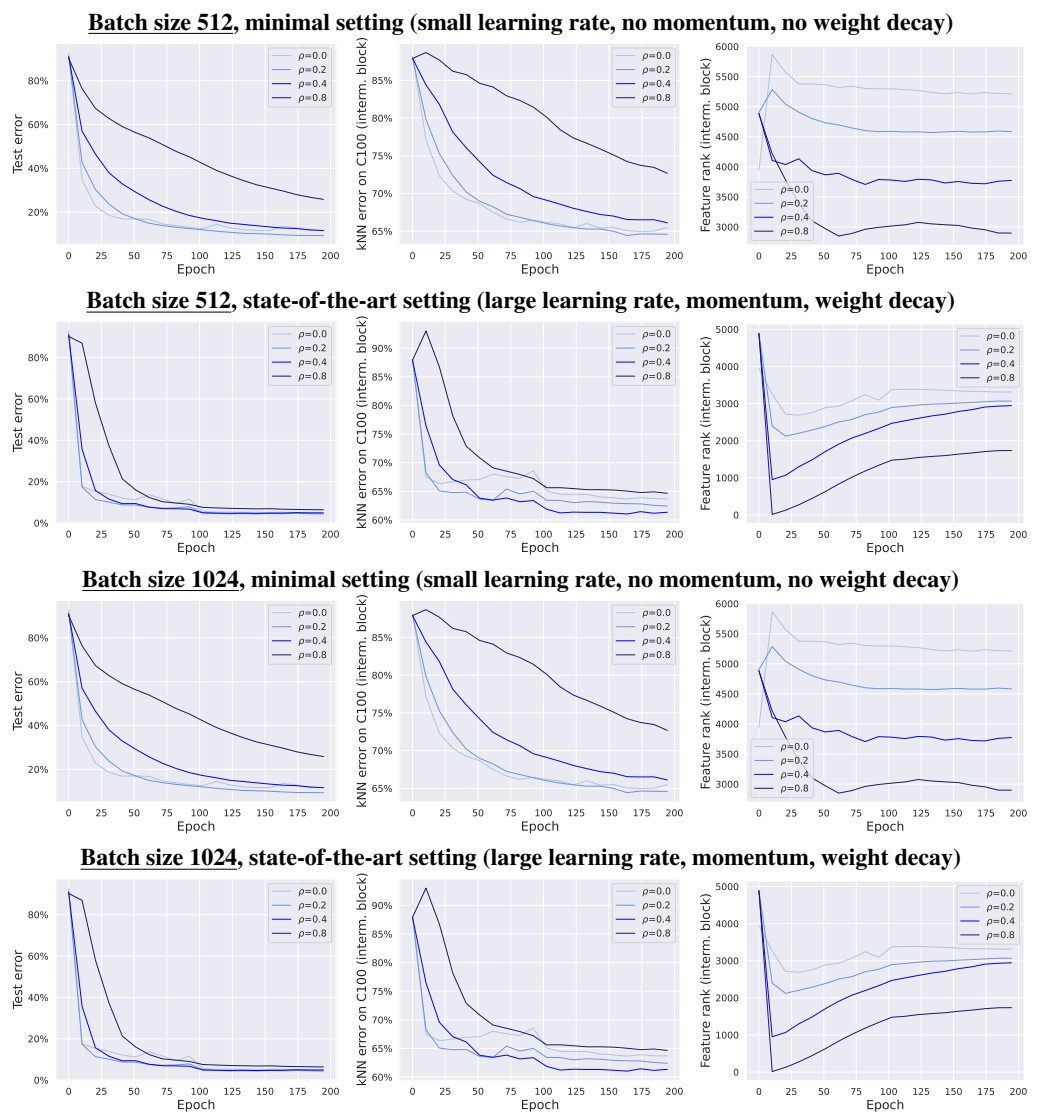

**Figure 11: ResNet-18 on CIFAR-10 with different batch sizes.** Similarly to the experiments in Figure 1 that were done with batch size 256, SAM with larger batch sizes (512 and 1024) also improves test error (*left*), leads to more generalizable features (*middle*), and noticeably reduces the feature rank at the intermediate ResNet block (*right*).

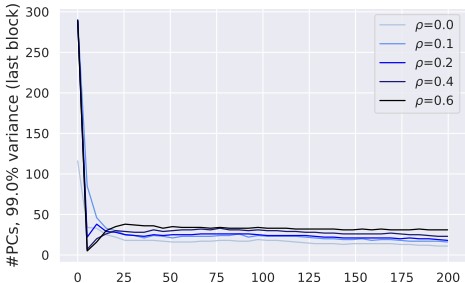

**Figure 12: ResNets-18 on CIFAR-10.** Feature ranks computed at the *last* block of ResNet (just before the linear head). We see that the neural collapse phenomenon occurs: the feature rank converges to 10 (i.e., the number of classes) even for $\rho = 0$. The feature rank with SAM is lower at the beginning but then increases to a slightly higher value than 10 towards the end.

## C  Additional results for contrastive image-text training

In Figure 13, we plot the retrieval error and feature rank for models with a *frozen* (i.e., not updated during training) BERT text encoder trained with different $\rho$ of SAM. We observe that the low-rank effect in the final text features (as well as image features) occurs even when a single layer is attached to a fixed BERT text encoder. Moreover, as a side note, SAM also improves the retrieval error in this setting as well.

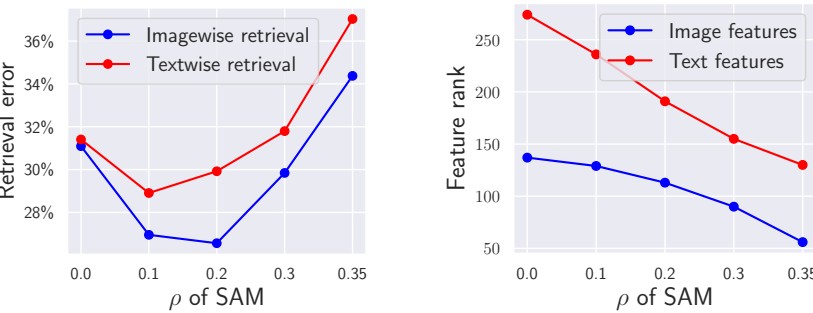

**Figure 13: Contrastive image-text training on MS-COCO (with a frozen text encoder).** The low-rank effect observed in the final text features even when a single layer is attached to a fixed BERT text encoder.

## D  Additional results for two-layer networks

In Figure 14, we plot the same metrics as in Figure 5 but, in addition, for tanh and absolute value activations. We can see that these models, when trained with higher $\rho$ of SAM, have a smaller feature rank, smaller number of active tanh/abs units, and higher weight norm. This illustrates the fact that the same low-rank mechanism can be at play for many activations, not just for ReLU.

In Figure 15, we confirm empirically that using only the first-order terms in the SAM update—which is equivalent to the gradient norm regularization $\sum_{i=1}^{n} \|\nabla \ell_i(\boldsymbol{\theta})\|_2$—leads to the same effect in all key metrics including the feature rank and the number of active ReLUs. This empirical validation supports the argument outlined in Proposition 1 which is based only on the first-order terms that empirically appear to be dominant for the low-rank effect of SAM.

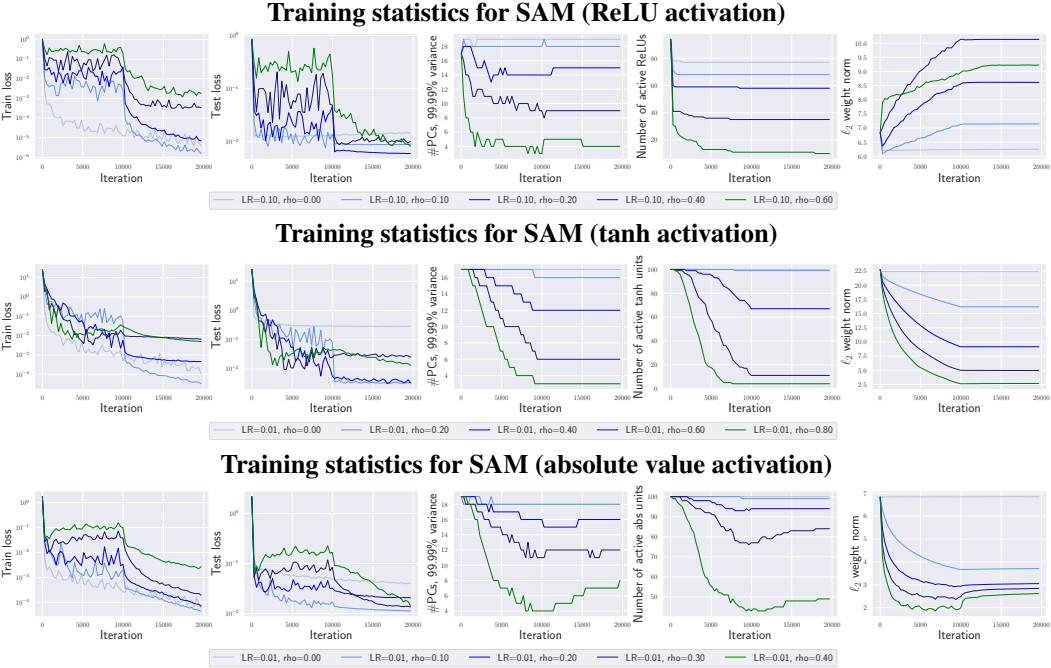

**Figure 14: Two-layer ReLU/tanh/abs networks in the teacher-student setup.** We can see that models trained with higher $\rho$ of SAM have a smaller feature rank, smaller number of active tanh/abs units, and higher weight norm.

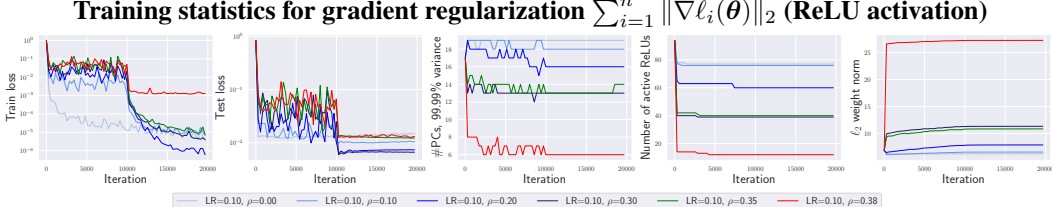

**Figure 15: Two-layer ReLU networks in the teacher-student setup.** We can see that models trained with higher $\rho$ of gradient regularization (which can be seen as the first-order approximation of SAM) have a smaller feature rank, smaller number of active ReLUs, and higher weight norm.

# E   Additional results for investigations of the low-rank mechanism on deep networks

In Figure 16, we show more detailed plots compared to Figure 7 regarding the behavior of feature ranks at different layers of the vision transformer on MS-COCO. First, we note that the feature rank grows over the residual blocks starting from very small values in the first blocks due to the identical positional encoding applied to all inputs at the first layer. Second, we do not observe a clear trend in the reduction of rank for larger $\rho$ after applying attention or GeLU activations *in the residual branches*. This is in contrast to the ranks in the *residual stream* (i.e., the values right after adding the identity) after attention subblocks which, as highlighted in Figure 7 in the main part, increase less for models with higher $\rho$. For the sake of completeness, the absolute values of ranks in the residual stream are shown in Figure 16.

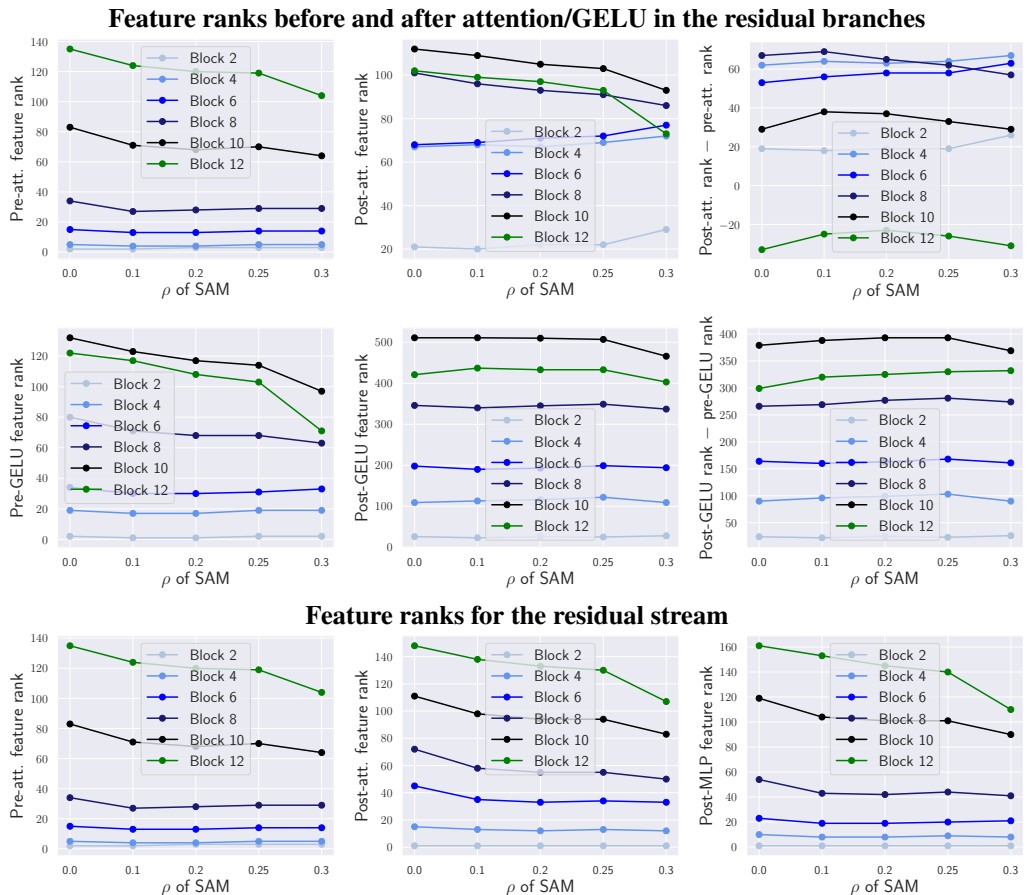

**Figure 16: Contrastive image-text training on MS-COCO (image encoder).** A more detailed look on the feature rank trend at different blocks depending on the $\rho$ of SAM.

Finally, in Figure 17, we show the feature ranks for the last layer of the text encoder with *unfrozen* and *frozen* text encoders. Note that a single linear layer is used on top of the frozen text features to match the dimensions of the image and text encoders within the same feature space. We conclude that even a single linear layer trained on top of the text encoder can be sufficient for the low-rank effect of SAM.

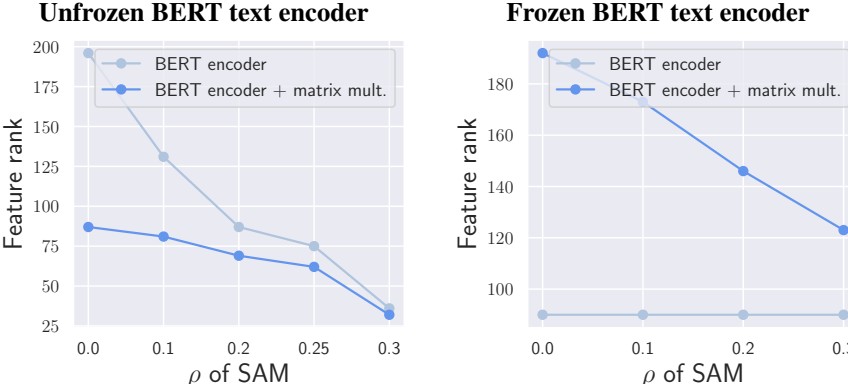

**Figure 17: Contrastive image-text training on MS-COCO.** We show the feature ranks for the last layer of the text encoder with *unfrozen* and *frozen* text encoders. Matrix multiplication denotes the last linear layer used to match the dimensions of the image and text encoders within the same feature space.

