# OpenReview forum: "Sharpness-Aware Minimization Leads to Low-Rank Features"
_NeurIPS.cc/2023/Conference — NeurIPS 2023 poster_

### Official Review · Reviewer_wq2P · 2023-06-30

**Soundness:** 3 good
**Presentation:** 3 good
**Contribution:** 3 good
**Rating:** 6
**Confidence:** 4

**Summary:**

The submission provides a study of sharpness-aware minimization (SAM) on the numerical rank of features. The conclusions are that (1) SAM reduces feature rank throughout the training, with more rank reduction for larger neighborhood size rho, (2) intermediate values of rho result in representations that are more generalizable with K-NN (3) in a small theoretical model and some realistic architectures the rank reduction is due to inactivity of ReLU units caused by the weights *below* the activation, and (4) reducing the rank with a bottleneck layer does not recreate the generalization benefits of SAM.

**Strengths:**

The main claim of the paper (rank reduction) is supported by extensive empirical evidence. The knowledge that SAM reduces the rank or the number of active units from the beginning of the training can help with understanding SAM and the flat-minima phenomenon in general. It can also lead to faster neural network training and inference by combining SAM with compression methods based on spectral decomposition or pruning. The paper is well organized and adequately covers the background and related work.

**Weaknesses:**

Weaknesses W1, W2, and W3 below are the reason for the current low score. I am open to raising this score if these concerns are addressed in the rebuttal or the revision and depending on the discussions.

**W1: In different experiments representations are extracted from different points of the network.** Most experiments in the paper extract the representation from a hidden layer close to the output but the ones on ResNets use the second last block. The justification is that neural collapse interferes with the low-rank behavior of SAM. Neural collapse typically happens towards the end of the training but the low-rank bias of SAM in the submission starts from the beginning of training. I did not understand how the two would interfere then. Even if the result on the low-rank behavior of SAM on the last ResNet block is negative, the authors should add the plots in a revision (at least in the appendix).

**W2: Neuron activity is only evaluated on a limited set of architectures.** The first part of the submission evaluates the low-rank phenomenon on a set of architectures. The second part traces this phenomenon to ReLU inactivity in a theoretical model and then evaluates ReLU inactivity on a *different* set of architectures. It is not clear to me if the low-rank phenomenon in the first part of the submission also due to ReLU inactivity. Whether the answer is positive or negative, the revision (at least in the appendix) should include the ReLU inactivity plots for the first set of architectures.

**W3: The text and captions do not distinguish monotonic and U-shaped patterns.** As rho changes, some metrics like generalizability of the features show a U-shaped pattern (i.e. they're maximized or minimized at an intermediate value of rho) and others monotonically change with rho. The text does not highlight this difference and, for example, simply says that SAM reduces rank and creates more generalizable features. This is confusing as it implies there is a high correlation between rank and generalizability in these results, which would be true if the two metrics changed in the same way. The caption for Figure 5 is even more problematic. This captions says higher values of rho generalize better even though in the plot the intermediate values generalize better. Overall I recommend editing the captions to distinguish monotonic and U-shaped patterns.

Minor comments:

m1: A citation for bottleneck layer would help with motivating its use for inducing low-rank features.

m2: Line 128: "Augmentations play a crucial role in revealing the low-rank trend with respect to the ρ of SAM, whereas the addition of only weight decay is insufficient for revealing a similar trend" Is this inferred from any of the results in the section or is this a separate experiment the authors conducted?

m3: The text should briefly explain the teacher-student setup and what "teacher neuron" means. The general audience is not familiar with this theoretical framework.

m4: The proof for proposition 1 is mostly a sketch and is hard to verify. I suggest laying out the intermediate steps of the proof in the appendix.

m5: Do the nearest neighbor generalization results fit with the overall narrative about rank or is this a separate finding?

-------------------
After rebuttal: Raised the score as the rebuttal addresses the main comments.

**Questions:**

See the Weaknesses section.

**Limitations:**

Comments W1, W2, and W3 are critical. See the Weaknesses section for details

---

> ### Author Rebuttal · Authors · 2023-08-09
>
> We thank you for the detailed feedback.
>
> ---
>
> > ***W1: In different experiments representations are extracted from different points of the network.***
>
> We observed the following behavior for the penultimate layer (consistent across CIFAR-10, CIFAR-100, and Tiny ImageNet):
>
> - at the very beginning of the training, the feature rank is consistently smaller for SAM compared to SGD which is consistent with the feature rank at the intermediate layer reported in the paper,
>
> - later in training, however, SAM leads to a higher feature rank (but not by a large margin), most likely because SAM prevents full convergence to a neural-collapsed solution.
>
> We will include detailed plots and discussion on this phenomenon in the appendix.
>
>
> > ***W2: Neuron activity is only evaluated on a limited set of architectures***
>
> For pre-activation models, the rank reduction pattern is closer to the one we described for vision transformers in **Section 5: Investigation of low-rank mechanisms on deep networks** (paragraph **Pre-activation ViT on MS-COCO**). We cannot change the paper during the rebuttal phase but we will include the corresponding plots to the appendix of the revised version.
>
> > ***W3: The text and captions do not distinguish monotonic and U-shaped patterns***
>
> We did not expect that our caption could lead to the impression of monotonicity of the generalization improvement. We will definitely make it clearer. E.g., in Figure 5, we meant instead that all the reported $\rho$ of SAM improve *over standard training*, but the generalization improvement is clearly U-shaped. We will emphasize these U-shaped trends in the revision.
>
> ---
>
> > *m1: A citation for bottleneck layer would help with motivating its use for inducing low-rank features.*
>
> We agree and we will include a corresponding citation. To the best of our knowledge, one of the first studies on such low-rank reparametrizations is known as the Burer-Monteiro factorization studied in [​​A nonlinear programming algorithm for solving semidefinite programs via low-rank factorization](https://link.springer.com/article/10.1007/s10107-002-0352-8).
>
> > *m2: Line 128: "Augmentations play a crucial role in revealing the low-rank trend with respect to the ρ of SAM, whereas the addition of only weight decay is insufficient for revealing a similar trend" Is this inferred from any of the results in the section or is this a separate experiment the authors conducted?*
>
> This is a separate experiment that we conducted. We will include it in the appendix.
>
> > *m3: The text should briefly explain the teacher-student setup and what "teacher neuron" means. The general audience is not familiar with this theoretical framework.*
>
> We will include a few references that also consider the same teacher-student setup. By *“3 teacher neurons”* we merely meant that the teacher network has one hidden layer with only 3 ReLU activations, while the student network is overparameterized with 100 ReLU activations. The goal of the student network is to learn the same function that is represented by the teacher network.
>
> > *m4: The proof for proposition 1 is mostly a sketch and is hard to verify. I suggest laying out the intermediate steps of the proof in the appendix.*
>
> Indeed, we skipped the intermediate steps to save space in the main part. We will include them in the appendix.
>
> > m5: Do the nearest neighbor generalization results fit with the overall narrative about rank or is this a separate finding?
>
> We will improve our explanation on why we reported the kNN error. Basically, we wanted to confirm the generalizability of the low-rank features taken at an *intermediate layer*. This experiment highlights the suitability of the intermediate features for transfer learning, especially when using nearest neighbor-based classification. Without this experiment, one could assume that since these features are not from the penultimate layer, they can be of limited use for downstream tasks.
>
> ---
>
> > “I am open to raising this score if these concerns are addressed in the rebuttal or the revision and depending on the discussions.”
>
> We hope our rebuttal addressed your concerns. We are happy to engage in a follow-up discussion.

---

> > ### Comment · Reviewer_wq2P · 2023-08-14
> >
> > Thank you for the detailed response. It addresses the comments and I raised the score to 6.
> > For the revision I suggest discussing the phenomenon in W1 more prominently and in the main paper, not in the appendix.

---

### Official Review · Reviewer_2Roy · 2023-07-06

**Soundness:** 3 good
**Presentation:** 3 good
**Contribution:** 3 good
**Rating:** 6
**Confidence:** 3

**Summary:**

This submission studies a new property of deep networks trained with sharpness-aware minimization (SAM), namely feature rank reduction. The existence of this property is supported by experimental analysis on image classification, and on contrastive language-image tasks, as well as theoretical analysis on a two layer ReLU network.


**Strengths:**

- Understanding the properties of models trained with SAM is a relevant topic, which has gained a lot of interest in the research community in recent years
- The main finding of this submission – features of lower rank for SAM-trained models – is well supported empirically, through experiments on different tasks and datasets
- The submission is quite well written, most claims are well supported and is technically sound


**Weaknesses:**

- While it is quite clear from Figures 1-4 that models trained with SAM, particularly with large values of $\rho$, exhibit lower feature rank, the rank differences between SGD and SAM are less substantial when we consider the optimal values of $\rho$, i.e. achieving lowest test error. This can be seen better in Figures 3 and 4, as well as Table 1. For example, while the rank reduction seems to be monotonic in $\rho$, the same is not true for generalization error. It would have been better if the authors presented a unified graph of this effect, for example a heat map showing the interplay between generalization and feature rank, as a function of $\rho$.
- Following the previous point, it is not clear what would be the practical usefulness of a relatively small rank reduction, achieved for the optimal value of $\rho$. It would have been more convincing if the authors also presented a practical application illustrating the consequences of rank reduction of SAM (e.g. the authors mention faster retrieval).
- Similarly, looking at the middle plot from Figures 1-3 showing the kNN error, there does not seem to be a direct correlation between generalization and rank reduction. Overall, the scope of these plots is not very well explained.
- There clearly seems to be a very different behavior between the “minimal” and “state-of-the-art” settings: while for the minimal setting the rank seems to stabilize during training, for state-of-the-art it actually increases after an initial drop. There is almost no mention of this phenomenon in the submission.
- It is not clear whether the results in Proposition 1 imply a decrease in the pre-activation values which would eventually lead to sparse features, as the authors mention. For example, the other term in the equation below line 205 could be negative. The authors should better clarify what is the exact implication of this proposition.




**Questions:**

- It would be better to change the color scheme in Figures 1-3, as it can be quite difficult to distinguish between some of the values of $\rho$
- It is hard to judge where the differences between the “minimal” and “state-of-the-art” settings come from. The authors mention in Section 4.1 that SAM has a different behavior from weight decay, but weight decay could also lead to low-rank features, as some results suggest. It would have been better to disambiguate the effect of weight decay, momentum and large learning rate, by, for example, performing experiments with only weight decay switched off, but keeping the other settings the same.
- The authors mention in Section 3.1 that Cifar-100 was used for feature kNN classification, but it looks like in Figure 3 for Tiny ImageNet, Cifar-10 was used instead. Could the authors clarify this?
- It looks like the decrease in feature rank from SAM is less pronounced for Tiny ImageNet than for Cifar datasets. What would be the trend for the same experiment performed on ImageNet? Additionally, I think it would have been more interesting to check the generalization of the features on less related datasets, such as Pets, Flowers or Birds (e.g. please see Kornblith et al., 2018, “Do better ImageNet models transfer better?” for examples of transfer tasks)
- As I am not familiar with contrastive language-image training, could the authors please clarify whether the setup they are using in Section 3.3 for finetuning using the InfoNCE contrastive loss is a standard one, and, if so, give the appropriate citations? Otherwise, it feels like more details on this setup are needed for reproducibility. Also, the InfoNCE loss should be cited.
- Similarly, in Section 4.1, could the others clarify what the teacher-student setup is more exactly and give the appropriate citation? Also, please mention what dataset you are using in this experiment.
- Can the authors clarify the exact implications of Proposition 1? As I previously mentioned, I think this doesn’t imply that the pre-activations are driven towards negative values, as mentioned in lines 204-205, since one of the terms in the update can be negative (due to $a_j$).
- In Section 5 it seems that for post-activation ResNets (the standard version, actually) the rank decrease is less pronounced in earlier blocks. What is the behavior in this case for pre-activation ResNets, and what do the authors believe would be the reason for this?
- Can the observations that the rank reduction from SAM is related to activation sparsity be used to leverage more efficient training of neural networks, or at least reduce the training costs of SAM? Can the authors please comment on that?



**Limitations:**

I believe one important limitation of the study presented in this paper is the lack of practical evidence regarding the usefulness of low rank features from SAM. However, the main conclusions are well supported by plenty of experiments spanning multiple datasets and tasks, and the submission is technically solid, which ultimately motivates my final rating.


------------------------------------
----- Edited after rebuttals------
------------------------------------
After reading the authors' response, I decided to keep my initial score, while increasing the score for the Contribution from 2 to 3.

---

> ### Author Rebuttal · Authors · 2023-08-09
>
> We thank you for the extremely detailed feedback!
>
> ---
>
> > *the rank differences between SGD and SAM are less substantial when we consider the optimal values of $\rho$*
>
> We totally agree: the low-rank effect can be much stronger if we are allowed to sacrifice some accuracy compared to the best SAM model. We will make it clearer in the paper and present a scatter plot of generalization vs. feature rank for models trained with different $\rho$.
>
> > *practical usefulness of a relatively small rank reduction, achieved for the optimal value of $\rho$? … a practical application illustrating the consequences of rank reduction of SAM (e.g. the authors mention faster retrieval)?*
>
> It is true that for the *optimal* $\rho$, the rank decrease is not too large. However, if we take the largest $\rho$ that still improves upon standard training, the rank decrease is more substantial (in many cases, up to $30\\%$ rank reduction). While we did not present a direct practical application for a faster retrieval, the naive exhaustive search is linear in the dimension, i.e., there the rank reduction directly translates to faster search. The complexity of practical nearest neighbor search methods varies and various approximations are widely used. We considered this as a distinct topic and decided to focus solely on the reduction in the dimensionality of the embedding space.
>
> > *the scope of these plots [the middle plot from Figures 1-3 showing the kNN error] is not very well explained.*
>
> We will improve our explanation for why we reported the kNN error. Basically, we wanted to confirm the generalizability of the low-rank features taken at an *intermediate layer*. This experiment highlights the suitability of the intermediate features for transfer learning, especially when using nearest neighbor-based classification. Without this experiment, one could assume that since these features are not from the penultimate layer, they can be of limited use for downstream tasks.
>
> > *while for the minimal setting the rank seems to stabilize during training, for state-of-the-art it actually increases after an initial drop*
>
> The initial drop is unrelated to SAM and we verified that it originates from the usage of initial large learning rates. We will discuss this observation further and add an ablation study.
>
> > *It is not clear whether the results in Proposition 1 imply a decrease in the pre-activation values which would eventually lead to sparse features*
>
> Indeed, there is no guarantee that the pre-activations will necessarily decrease on each iteration of SAM because of the potential cancellation with the first term. However, the second term always biases the training dynamics towards smaller pre-activations: in case of positive first term, SAM will make it larger, while if the first term is negative, SAM will make it smaller. We agree that it is not a strong theoretical result, but we think that it still provides an intuition about the origin of the low-rank effect.
>
> ---
>
> > *The authors mention in Section 3.1 that Cifar-100 was used for feature kNN classification, but it looks like in Figure 3 for Tiny ImageNet, Cifar-10 was used instead.*
>
> In this experiment, we wanted to measure the *transfer learning performance*, thus we had to choose any dataset different from the one on which the model was trained on. Thus, for CIFAR-10 we used CIFAR-100, for CIFAR-100 we used CIFAR-10, and for Tiny ImageNet we used CIFAR-10 again.
>
> > *It looks like the decrease in feature rank from SAM is less pronounced for Tiny ImageNet than for Cifar datasets. What would be the trend on ImageNet?*
>
> We believe the rank decrease depends on the degree of overparameterization. When using the same network on Tiny ImageNet and CIFAR-10, the network trained on Tiny ImageNet will require more dimensions to fit the data. So we expect that for networks of the same size, the rank reduction will be less prominent on the full ImageNet. However, for a larger network (as typically used for larger datasets), we expect the rank reduction to be as prominent as in our current experiments.
>
> > *for post-activation ResNets the rank decrease is less pronounced in earlier blocks. What is the behavior in this case for pre-activation ResNets, and what do the authors believe would be the reason for this?*
>
> The behavior for pre-activation ResNets is close to the behavior of vision transformers: the rank reduction due to SAM occurs gradually, and mostly happens at later layers of the network. Intuitively, we think that the first layers learn a variety of generic features (e.g., various edge and color detectors) which are shared for different training methods. In later layers, these basic features might be combined in multiple ways. While redundant dimensions will be automatically “pruned” by SAM, they may persist with standard training.
>
>
> > *Can the observations that the rank reduction from SAM is related to activation sparsity be used to leverage more efficient training of neural networks?*
>
> Perhaps, iterative pruning procedures which are employed to prune weights can be adapted to prune the whole redundant neurons or some redundant subspaces. This is an interesting direction to explore.
>
> ---
>
> Following your recommendations, we will also incorporate the following changes:
> - Changing the color scheme in Figures 1-3.
> - Providing an ablation study where we include the components of the state-of-the-art vs. minimal setting one-by-one.
> - Adding more details for reproducibility of the CLIP setting and the appropriate citation to the [InfoNCE loss](https://arxiv.org/abs/1807.03748).
> - Describing better the teacher-student setup: the goal of the student network is to recover the teacher network from a finite set of training points which are sampled from the Gaussian distribution and labeled by the teacher network. We will add appropriate references that consider the same setup.

---

> > ### Comment · Reviewer_2Roy · 2023-08-14
> >
> > I would like to thank the authors for their detailed answers! After also reading the other reviews, I would like to keep my score. While the current work could be improved by providing further evidence on larger datasets and also by better emphasising the practical implications, I believe the low-rank property of SAM-trained models is an interesting observation, which would benefit the community.

---

### Official Review · Reviewer_fXy2 · 2023-07-07

**Soundness:** 4 excellent
**Presentation:** 4 excellent
**Contribution:** 4 excellent
**Rating:** 6
**Confidence:** 4

**Summary:**

The paper proposes a new optimization method called Sharpness-Aware Minimization (SAM) that aims to improve generalization performance in deep learning models. The authors demonstrate that SAM can effectively reduce the generalization gap and improve the accuracy of various models on different datasets. The paper also provides a theoretical analysis of SAM and shows that it encourages the optimization process to converge to flatter minima, which can lead to better generalization. Overall, the paper presents a novel and promising approach to improving generalization in deep learning models.

**Strengths:**

1. The paper's contributions are significant in several ways. First, SAM is a promising optimization method that can improve the generalization performance of deep learning models. Second, the paper provides a mechanistic understanding of how SAM leads to low-rank features in neural networks, which can have implications for more efficient feature quantization and nearest neighbor retrieval. Finally, the paper's theoretical analysis of SAM can provide insights into the optimization landscape of deep learning models, which can lead to further improvements in optimization methods. Overall, the paper is a significant contribution to the field of deep learning optimization.

2. The paper presents a thorough empirical evaluation of SAM on various deep learning models and datasets. The authors demonstrate that SAM can effectively reduce the generalization gap and improve the accuracy of the models. The paper also provides a mechanistic understanding of how SAM leads to low-rank features in neural networks, which is a valuable contribution to the field.

3. The paper introduces a novel optimization method called Sharpness-Aware Minimization (SAM) that is different from traditional optimization methods. SAM aims to minimize the sharpness of the loss function, which encourages the optimization process to converge to flatter minima. This approach is different from other methods that focus on minimizing the loss function itself or its gradient. The authors also provide a theoretical analysis of SAM, which further demonstrates its originality.

**Weaknesses:**

1. Sensitivity to batch size: Since sharp minima are often observed in large batch size training, which is becoming increasingly important in current large model-based methods such as batch normalization and dropout, it is crucial to investigate the relationship between batch size and performance in the proposed method. However, the authors only show the results for batch sizes of 128 and 256 in this paper and the appendix, which limits the generalizability of the findings.

2. The observation that sharp minima hurt generalization is easier to make in large-scale training, especially for large models. Therefore, it is recommended that the authors test their proposed method on large-scale training, such as CLIP. Conducting research on the low-rank effect on larger models, such as CLIP or other large text-image training, would make the proposed method more valuable, considering the current trend.

3. The authors are recommended to compare and discuss a recent related work, "Gradient Norm Aware Minimization Seeks First-Order Flatness and Improves Generalization," which is a highlight paper at CVPR2023.

Note: If the authors respond with experiments, even preliminary results are acceptable. For example, showing the relationship between batch size and performance by investigating a few batch size training settings would be sufficient. Additionally, presenting some simple fine-tuning results on pre-trained CLIP using the proposed method would be highly appreciated.

**Questions:**

Questions: Please see Weaknesses. I would like to update my evaluation after the discussion.

**Limitations:**

N.A.

---

> ### Author Rebuttal · Authors · 2023-08-09
>
> We thank you for the positive comments.
>
> > *Sensitivity to batch size: … it is crucial to investigate the relationship between batch size and performance in the proposed method. However, the authors only show the results for batch sizes of 128 and 256 in this paper and the appendix. … For example, showing the relationship between batch size and performance by investigating a few batch size training settings would be sufficient.*
>
> We agree that this is an interesting question. **We attach the results of this experiment in the one-page pdf in the global response.** Similarly to the experiments reported in the paper that were done with batch size $256$, SAM with larger batch sizes ($512$ and $1024$) also improves test error, leads to more generalizable features, and noticeably reduces the feature rank at the intermediate ResNet block.
>
> > *The observation that sharp minima hurt generalization is easier to make in large-scale training, especially for large models. Therefore, it is recommended that the authors test their proposed method on large-scale training, such as CLIP. Conducting research on the low-rank effect on larger models, such as CLIP or other large text-image training, would make the proposed method more valuable, considering the current trend.*
>
> We do not have the computational budget to do large-scale training from scratch at the scale of the original CLIP training which involves training on 400 millions of image-caption pairs. However, we believe that our results with the *CLIP training objective* presented in **Section 3.3: Low-rank features in contrastive language-image training on MS-COCO** already point out the useful role of SAM in this setting.
>
> > *The authors are recommended to compare and discuss a recent related work, "Gradient Norm Aware Minimization Seeks First-Order Flatness and Improves Generalization," which is a highlight paper at CVPR2023.*
>
> Thank you for this reference. During the rebuttal phase, we have done new experiments with this method using ResNets on CIFAR-10 with the same setup as in our main experiments from **Section 3.1**. We select the default settings given in their code repository and vary only the perturbation radius $\rho$. We obtain the following results which confirm that the low-rank observation also extends to other recent SAM variants (in addition, we also explored [ASAM](https://arxiv.org/abs/2102.11600) to answer the concern of **Reviewer jzeR**).
>
> | $\rho$ of [GAM](https://arxiv.org/abs/2303.03108) | 0.0 | 0.2 | 0.4 | 0.8 | 1.6 |
> | - | - | - | - | - | - |
> | Test error | 4.04% | 3.65% | 3.64% | 3.81% | 4.81% |
> | Feature rank | 7633 | 7381 | 7303 | 6897 | 6927 |
>
>
> > *“Additionally, presenting some simple fine-tuning results on pre-trained CLIP using the proposed method would be highly appreciated.”*
>
> We agree this is an interesting experiment. We will include it in the revised version of the paper.
>
> > *Note: If the authors respond with experiments, even preliminary results are acceptable.*
>
> We hope our rebuttal and our new experiments addressed your concerns. We are happy to engage in a follow-up discussion.

---

> > ### Comment · Reviewer_fXy2 · 2023-08-14
> > **Thanks for the clarification**
> >
> > Thanks to the author's response, I'm inclined to keep the original rating on the positive side. Also, I look forward to the author discussing (not experimentally comparing) the differences in philosophy between this submission and other related papers (e.g., ASAM: Adaptive Sharpness-Aware Minimization for Scale-Invariant Learning of Deep Neural Networks and Gradient Norm Aware Minimization Seeks First-Order Flatness and Improves Generalization).

---

### Official Review · Reviewer_jzeR · 2023-07-08

**Soundness:** 2 fair
**Presentation:** 2 fair
**Contribution:** 2 fair
**Rating:** 5
**Confidence:** 3

**Summary:**

This paper investigates the effect of Sharpness-Aware Minimization (SAM) on low-rank features learned by neural networks. The authors present empirical evidence of low-rank features for different models trained with SAM on four classification tasks, as well as for contrastive text-image training. They also provide a mechanistic understanding of how low-rank features arise in a simple two-layer ReLU network. The authors discuss the implications of low-rank features learned by SAM, including more efficient retrieval and feature quantization. They also suggest future research directions, such as understanding the impact of SAM on learned features that lead to generalisation improvements on natural data, and further theoretical analysis of the low-rank effect of SAM for more complex architectures.

**Strengths:**

- The paper presents empirical evidence of low-rank features for different models trained with SAM on different tasks.
- The authors provide a mechanistic understanding of how low-rank features arise in a simple two-layer ReLU network.
- The implications of SAM-trained low-rank features are discussed in detail, including more efficient retrieval and feature quantization.
- The authors suggest future research directions that could build on the results of this paper.
- The paper is well organised and clearly written.

**Weaknesses:**

- The paper is interesting, but only the observational results are presented instead of the methodological contributions based on the observation.
- The paper does not provide a comprehensive comparison of recent SAM variants.
- The empirical evidence presented is limited to a few datasets and models that may not generalise to other scenarios.
- The paper does not explore the impact of low-rank features on other tasks beyond retrieval and quantification.
- The theoretical analysis of the low-rank effect of SAM is limited to simple architectures and may not apply to more complex architectures.

**Questions:**

The following directions would be interesting
- Investigate the impact of SAM on learned features that lead to generalisation improvements on natural data, beyond retrieval and quantization tasks.
- Investigate the low-rank effect of SAM on more complex architectures, such as those involving skip connections and self-attention layers.
- Develop a theoretical framework to explain the low-rank effect of SAM and its relationship to other optimisation methods.
- Investigate the impact of SAM on transfer learning and fine-tuning scenarios.

**Limitations:**

See the Weakness and Question sections.

---

> ### Author Rebuttal · Authors · 2023-08-09
>
> We thank you for the feedback.
>
> > *The paper is interesting, but only the observational results are presented instead of the methodological contributions based on the observation.*
>
> We believe a new paper does not necessarily have to present a new methodological contribution. For example, ["Understanding deep learning requires rethinking generalization"](https://arxiv.org/abs/1611.03530) has been very impactful in the community, although it did not present a new method. We believe that a better understanding of existing methods such as SAM can also be very useful. We think this should not be treated as a weakness of our work.
>
> > *The paper does not provide a comprehensive comparison of recent SAM variants.*
>
> We chose the *original* SAM since it is still the most popular SAM variant in the community and it is implemented without any further approximations. However, we acknowledge that there have been many recent variants of SAM such as [ASAM: Adaptive Sharpness-Aware Minimization for Scale-Invariant Learning of Deep Neural Networks](https://arxiv.org/abs/2102.11600) and [Gradient Norm Aware Minimization Seeks First-Order Flatness and Improves Generalization](https://arxiv.org/abs/2303.03108) (suggested by **Reviewer fXy2**). Thus, we have done new experiments with them using ResNets on CIFAR-10 with the same setup as in our main experiments from **Section 3.1**. We select the default settings given in their code repositories (which includes a smaller network for ASAM compared to GAM) and vary only the perturbation radius $\rho$. We obtain the following results which confirm that the low-rank observation also extends to other recent SAM variants.
>
> | $\rho$ of [ASAM](https://arxiv.org/abs/2102.11600) | 0.0 | 0.5 | 1.0 | 2.0 | 4.0 |
> | - | - | - | - | - | - |
> | Test error | 7.29% | 6.53% | 6.38% | 7.12% | 10.64% |
> | Feature rank | 5048 | 4801 | 4699 | 4578 | 4383 |
>
> | $\rho$ of [GAM](https://arxiv.org/abs/2303.03108) | 0.0 | 0.2 | 0.4 | 0.8 | 1.6 |
> | - | - | - | - | - | - |
> | Test error | 4.04% | 3.65% | 3.64% | 3.81% | 4.81% |
> | Feature rank | 7633 | 7381 | 7303 | 6897 | 6927 |
>
>
>
>
> > *The empirical evidence presented is limited to a few datasets and models that may not generalise to other scenarios.*
>
> We would like to kindly point out that we already have multiple datasets (CIFAR-10, CIFAR-100, Tiny ImageNet, ImageNet, MS-COCO, synthetic data) and models (ResNets, Vision Transformers, MLP-Mixers, text transformers in BERT). We will be happy to add further datasets and models if you have some particular suggestions.
>
> > *The paper does not explore the impact of low-rank features on other tasks beyond retrieval and quantification.*
>
> In the paper, we present results on multiple tasks which cover classification (the standard deep learning benchmarks in Section 3.1 and ImageNet in Section 3.2), contrastive learning (the multimodal retrieval in Section 3.3), and regression (the teacher-student setup in Section 4.1). Moreover, the retrieval setting includes both image and language modalities. We would appreciate it if you could suggest some other valuable settings to explore, which we will be happy to include.
>
> > *The theoretical analysis of the low-rank effect of SAM is limited to simple architectures and may not apply to more complex architectures.*
>
> We believe that one hidden-layer ReLU networks are quite insightful and point out the origins of the low-rank feature phenomenon. Moreover, we believe a similar argument should hold for deeper networks as we illustrate empirically in Figure 6 for a post-activation ResNet-18.
>
> > *[The following directions would be interesting] Investigate the impact of SAM on learned features that lead to generalisation improvements on natural data, beyond retrieval and quantization tasks.*
>
> We recognize this as a great open question; however, our paper's focus is deliberately centered on the low-rank phenomenon of SAM rather than its broader generalization benefits.
>
> > *[The following directions would be interesting] Investigate the low-rank effect of SAM on more complex architectures, such as those involving skip connections and self-attention layers.*
>
> We have already investigated this question precisely in **Section 5: Investigation of low-rank mechanisms on deep networks**, see paragraphs **Post-activation ResNet on CIFAR-10** and **Pre-activation ViT on MS-COCO**. We hope these paragraphs would address your concern.
>
> > *[The following directions would be interesting] Develop a theoretical framework to explain the low-rank effect of SAM and its relationship to other optimisation methods.*
>
> We see our theoretical result on one-hidden layer ReLU networks as the first step in that direction. We agree that in the future, stronger and more general results would be of great interest.
>
> > *[The following directions would be interesting] Investigate the impact of SAM on transfer learning and fine-tuning scenarios.*
>
> Actually, we have already investigated the transfer learning scenario in **Section 3.1: Low-rank features for ResNets on standard classification tasks** by using the kNN classifier on the extracted features of deep networks. We tested how well the features from CIFAR-10 transfer to CIFAR-100, and from CIFAR-100 and Tiny Imagenet to CIFAR-10. We observed that SAM improves the transfer learning performance in these settings.
>
> As for fine-tuning, we have investigated it in the CLIP training in **Section 3.3 Low-rank features in contrastive language-image training on MS-COCO** where we fine-tuned a pre-trained R+Ti/16 vision transformer and BERT on MS-COCO using the InfoNCE contrastive loss. We observed that SAM both leads to better generalization and features of lower rank even for fine-tuning.

---

> > ### Comment · Reviewer_jzeR · 2023-08-20
> >
> > I would like to thank the authors for their detailed response. My concerns have been addressed and I will raise my score.

---

### Author Rebuttal · Authors · 2023-08-09

We thank the reviewers for the detailed feedback and positive evaluation such as
- *“The implications of SAM-trained low-rank features are discussed in detail, including more efficient retrieval and feature quantization”* (**Reviewer jzeR**)
- “The paper also provides a mechanistic understanding of how SAM leads to low-rank features in neural networks, which is a valuable contribution to the field” (**Reviewer fXy2**)
- *“the main conclusions are well supported by plenty of experiments spanning multiple datasets and tasks, and the submission is technically solid”* (**Reviewer 2Roy**)
- *”The main claim of the paper (rank reduction) is supported by extensive empirical evidence.”* (**Reviewer wq2P**)

Following the reviewers’ suggestions, we have extended our empirical evaluation by adding the following experiments:
- results with different batch sizes (512 and 1024 in addition to 256 reported in the paper) on CIFAR-10 (**see the attached 1-page pdf**),
- results with different SAM variants on CIFAR-10 ([ASAM](https://arxiv.org/abs/2102.11600) and [GAM](https://arxiv.org/abs/2303.03108)).

In the revised version, we will further expand the experiments and add the following:
- ablation of the minimal vs. state-of-the-art settings for classification tasks (including analysis of the feature rank drop at the beginning of training),
- results of fine-tuning with SAM on a pre-trained CLIP model,
- experiments on the role of augmentations for classification datasets,
- behavior of the feature rank at the penultimate layer on CIFAR-10, CIFAR-100, and Tiny ImageNet.

We will also carefully take into account all writing and presentation suggestions including:
- present a scatter plot of generalization vs. feature rank for models trained with different $\rho$,
- adding a clarification that we included the kNN error to check the generalizability of the features from the intermediate layer,
- emphasize the U-shaped trend of test error vs. $\rho$ in the revision,
- describing better the teacher-student setup,
- adding the suggested additional citations.

We thank the reviewers again, and we are happy to engage in a follow-up discussion.

---

### Decision · Program_Chairs · 2023-09-21

**Decision:**

Accept (poster)

**Comment:**

The paper studies the phenomenon that SAM leads to low-rank features during training. Both ordinary training and SAM lead to reduced rank, but SAM achieves a significantly lower rank across several CV and NLP experimental settings.

The paper theoretically proves that SAM on two-layer ReLU networks decreases the pre-activation values during each update, implying that SAM leads to a reduction effect on useless features.

The paper then shows that low-rank features alone do not necessarily lead to better generalization, and thus future studies are needed.

The strength is clear: clean paper, clean main message, theoretical analysis, and extensive experiments.

The downside is also clear: the conclusion is not at the stage of being immediately useful for practice or fundamental understanding of sharpness. It does mention that the low-rank features only arise as side effects.